# Power production and area usage of offshore wind and the relationship with available energy in the atmosphere

Ole Anders Nøst[ID]*

Oceanbox AS, Krambugata 2, Trondheim, Norway

* ole.anders.nost@oceanbox.io

## Abstract

This paper presents an analysis of the area dependency of power and capacity density of wind farms, based on derivations of the available energy in the atmosphere and data on the power production of existing wind farms in the North Sea. The amount of available energy is determined by the mechanical energy budget of the atmospheric boundary layer and is found to be a function of $C/A$, where $C$ is the circumference and $A$ is the area of the wind farm. The actual power production of 31 wind farms is analyzed, and the power production in 23 of the largest wind farms follows the same functional form as the available energy, indicating that the power production in these wind farms is limited by the available energy. The power density in the North Sea, as a function of the area usage, is found by fitting the actual power density of existing wind farms to the expression for available energy in the atmospheric boundary layer. Wind farms below 10 $km^2$ can produce more than 6 $Wm^{-2}$, but the power density rapidly decreases with area. Wind farms with an area of about 1000 $km^2$ will produce $\sim 1$ $Wm^{-2}$, and power densities will asymptotically approach a value of $0.78 \pm 0.58$ $Wm^{-2}$ for increasing wind farm area. Since atmospheric energy input is the limiting factor for power production, an atmospheric energy budget is vital for a reliable estimate of offshore wind power potential.

**Data availability statement:** The installed capacity, area and capacity factors for all wind farms are given in the Supporting Information files. Half-hourly power production for all wind farms in the study can be downloaded from

## Introduction

In replacement of fossil fuels, wind energy is thought to be a main source of clean energy, providing up to half of global electricity needs [1]. In the Ostend declaration, the energy ministers of nine European countries have agreed on a large scale-up of offshore wind in the North Sea. The goal is to install 120 GW by 2030 and 300 GW by 2050. Today's offshore wind capacity in the North Sea is 35 GW according to Wind Europe [2]. Such a large increase in installed offshore wind capacity will have consequences for the environment, the area of power production, and the regulations of offshore wind development. A central topic is wind wakes created by offshore wind farms.

Wind farms extract kinetic energy from the wind, resulting in a reduced wind speed inside the wind farm and in its wake. Wind farm wakes can be seen and analyzed from satellites. Djath et al. [3] analyzed SAR images and found that the wakes could reach more than 70 km

https://doi.org/10.5061/dryad.w6m905qzk. The atmospheric data needed to do the calculations of conversion from potential to kinetic energy within the atmospheric boundary layer and replicate the plot shown in Fig 6, can be downloaded from https://psl.noaa.gov/data/gridded/data.ncep.reanalysis2.html. More direct links to the exact files needed are given below. The data that is necessary to replicate the calculations in the manuscript are: Air temperature on pressure levels: datafile = air.YYYY.nc. Geopotential height on pressure levels: datafile = hgt.YYYY.nc. Omega (vertical velocity in pressure coordinates on pressure) levels: datafile = omega.YYYY.nc. Eastward wind component at pressure levels: datafile = uwnd.YYYY.nc. Northward wind component at pressure levels: datafile = vwnd.YYYY.nc. In the calculations in the manuscript the years 2020, 2021 and 2022 are used. In the datafile name YYYY is replaced by the year. The data on pressure levels can be downloaded from https://psl.noaa.gov/thredds/catalog/Datasets/ncep.reanalysis2/pressure/catalog.html. In addition to the data on pressure levels, the surface pressure is also needed. Pressure at surface: datafile = pres.sfc.YYYY.nc. Surface pressure data can be downloaded from https://psl.noaa.gov/thredds/catalog/Datasets/ncep.reanalysis2/surface/catalog.html.

**Funding:** This study was supported by the Norwegian Seafood Research Fund (https://www.fhf.no/) grant number 901916. The funders had no role in study design, data collection and analysis, decision to publish, or preparation of the manuscript.

**Competing interests:** The authors have declared that no competing interests exist.

downstream of the wind farm. Models show that for large wind farms, the wakes can reach 100 km downstream [4]. Wakes reduce the efficiency of power production and due to this, much work has been done to understand the most efficient layout of turbines in a wind farm. Detailed wake models are important for this work, and a recent overview of wake models is given by [5]. Wakes typically reach farther downstream under stable atmospheric conditions [3,6], because atmospheric stability reduces turbulent exchange of kinetic energy from above.

A main effect of wind farm wakes is the induction of vertical movement in the ocean [7,8]. Vertical movement is driven by Ekman dynamics resulting from the reduction in wind stress within the wind farm and the wake [7]. The vertical movement is characterized by alternating up and downwelling patterns that form dipoles in sea surface elevation [9]. Changes in stratification are also shown to be an effect of the extraction of kinetic energy from the wind [8,10], and models also show reduced ocean current velocities due to reduced surface wind stress [10].

Using a numerical ecosystem model, Daewel et al. [10] show that wind wakes lead to changes in primary production. They see a reduction in primary production at the wind farm sites and an increase in the surrounding regions, leading to a change in the spatial distribution. In addition, changes in stratification and a reduction in vertical mixing due to reduced wind lead to decreased oxygen levels [10], which again has an effect on the ecosystem [11]. More information on the impact on the marine ecosystem of offshore wind farms is provided by Wang et al. [11].

The large planned increase in offshore wind in the North Sea will lead to an increase in wake effects. New wind farms can reduce power production in existing wind farms, which can lead to legal conflicts between wind farm operators. When wakes extend across national borders, conflicts between states may also occur. This topic is introduced by Finserås et al. [12], who explored the case of Sørlige Nordsjø II, an offshore region in the Norwegian economic zone that is opened for the development of offshore wind farms. Finserås et al. show that the wakes from the Sørlige Nordsjø II will regularly stretch into the Danish economic zone. This and similar cases may lead to conflicts that slow the development of offshore wind.

The planned scale-up of offshore wind in the North Sea will lead to a massive increase in wind farm wakes, which will reduce the area efficiency of power production [13]. To investigate the environmental effect of an offshore wind farm, we need to know the required area. Here, the area of a wind farm is defined as the area within the outer turbines. For a given installed capacity, the area depends on the density of the installed turbines. The capacity density is the installed capacity per area given in $Wm^{-2}$. If the turbines are installed too dense, the capacity factor will decrease [14]. The capacity factor represents the power production as a percentage of the installed capacity. The power density is the power production per area given in $Wm^{-2}$. The installed capacity should be adjusted to the estimated power density in order to maximize the capacity factor. The area needed for the installation of a given total capacity therefore depends on the expected power density. The area is also a vital parameter to predict the environmental effects caused by wind wakes. Thus, accurate estimates of power density are highly important, both for the most efficient layout of turbines and for predicting environmental consequences.

Despite the plans for scale-up, there are large disparities in estimates of power potentials. This is clearly documented in Pelser et al. [15]. How much power can offshore wind produce per area? The large variations in the estimates of the power potential show that we do not know the answer to this question accurately. This means that we do not have an accurate estimate of the future area needed for offshore wind. The area usage is the main parameter to understand how a scale-up of offshore wind will influence the environment and how offshore wind can co-exist with fisheries. The motivation for the work presented in this paper is

to improve estimates of the power potential and area usage of offshore wind in the North Sea, by describing the first-order physics responsible for the energy input to wind farms.

A common approach for wind resource assessments is to estimate the available wind power from global reanalysis wind speed datasets together with a detailed analysis of the areas available for wind power [16–19]. Eurek et al [16] use a turbine distance of 10 rotor diameters, corresponding to a capacity density of 5 $Wm^{-2}$, and accounts for 5% loss due to maintenance and 10 % wake loss. Bosch et al [18] also use a turbine distance of 10 rotor diameters, but assume 3 % maintenance loss and 11.45 % wake loss. Wake loss is derived from array efficiency taken from Gustavson [20]. The turbine spacing corresponds to a capacity density of 3.14 $Wm^{-2}$. Both Eurek et al [16] and Bosch et al [18] calculate capacity factors from local wind speed and turbine power curves, taking into account the assumed losses. Enevoldsen et al [19] assume an installation of $52 \cdot 10^6 MW$ over an area of about $5 \cdot 10^6 km^2$ in Europe, which corresponds to a capacity density of more than 10 $Wm^{-2}$. For calculating power production, they assume a 30 % capacity factor without discussing wake loss.

Pelser et al. [15] pointed out that simplistic methods to characterize wake losses were one of the causes for the large differences in power potential estimates. The three examples of resource assessments described above [16,18,19] are an example of this. They operate with very different capacity densities, and wake losses and capacity factors are independent of the capacity density. Using data from wind farms in the USA, Miller and Keith [14] show that capacity factors decrease with increasing capacity density. Simulating wind farms in the North Sea, Akhtar et al. [13] show that capacity factors are reduced in large wind farms and in neighboring downwind farms by 20 % or more due to wake losses. To find how the capacity factors depend on the capacity density, a budget of mechanical energy is needed within the wind farm.

The power produced by a wind farm is the result of a balance between the energy input from the atmosphere to the wind farm area, frictional dissipation, and power production. The energy input to small wind farms is mainly due to incoming horizontal winds. For larger wind farms, the incoming horizontal winds are less important because the energy they carry powers the turbines at the boundary and does not reach the center of the wind farm. For very large wind farms, the incoming horizontal winds are negligible, and to produce electric power, the winds must be accelerated within the area of the wind farm. Miller and Kleidon [21], Kleidon and Miller [22] and Kleidon [23] assume that it is a turbulent kinetic energy flux from above that is the main supply of energy to a large wind farm, which is in agreement with the main view on large-scale wind farms [24,25]. Antonini and Caldeira [26] present an alternative to this main view and suggest that the main energy supply to large wind farms is the work done by the pressure gradient force on the wind. Frictional forces deviates the wind from geostrophy, leading to a wind component towards lower pressure, which is equivalent to a conversion from potential to kinetic energy. For the atmospheric boundary layer (ABL) as a whole, turbulent energy fluxes from above cannot be the main energy supply because turbulent fluxes are negligible above the boundary layer [27]. In a simple steady-state Ekman model of the boundary layer, the loss to frictional dissipation is balanced by the work done by pressure forces. Thus, since pressure forces are important for the energetics of the ABL, it seems logical that they are also important for the energy input to large wind farms.

Understanding the transition from small to large wind farms is important as a scale-up of offshore wind will involve larger wind farms. However, it is not clear from the scientific literature what size a wind farm must be to be classified as large. Stevens and Meneveau [24] divert between large (but not continent-wide) wind farms and extremely large (e.g., continent-wide) wind farms. What size can be classified as large is not given, but the comparison to continent-wide size hints at something very much larger than the wind farms existing today. A recent

study by Antonini and Caldeira [28] represents a wind farm as a change in surface roughness and looks at how an Ekman layer responds when flowing into the area of the wind farm. After traveling a length proportional to the geostrophic wind speed divided by the Coriolis parameter, the Ekman layer will be fully adjusted to the new surroundings. The length scale is about 30 km for a wind speed of 8 $ms^{-1}$ and a Coriolis parameter of $1.05 \cdot 10^{-4} s^{-1}$. This is in line with the observation-based study by Miller and Keith [14] which showed a clear reduction in power densities with wind farm area, up to about 100 $km^2$, where wind farms larger than this produced about 0.5 $Wm^{-2}$.

A mechanical energy budget for a wind farm is needed to properly account for wind wakes in a resource assessment. Such energy budgets are used by Miller and Kleidon [21] and Kleidon and Miller [22]. They find an upper limit of power density at $\sim 1\ Wm^{-2}$ for large wind farms. This is in reasonable agreement with the mechanical energy cycle of the atmosphere, where, on a global average, about 2.5 $Wm^{-2}$ is converted from potential to kinetic energy [29–34]. However, the power density in large wind farms can vary with location and wind resources. Volker et al [35] simulate wind farms of different sizes in areas with different wind conditions. They find that very large wind farms ($10^5 km^2$) can have power densities that exceed 3 $Wm^{-2}$ in areas with favorable wind conditions. Disagreements about how to estimate the power production of very large wind farms are highlighted in a discussion between Badger and Volker [36] and Miller and Kleidon [37].

Do wind farms need to be continent-wide to reach the limit given by the atmospheric energy input, or is this limit already reached in existing wind farms? Since capacity densities are typically larger than the 2.5 $Wm^{-2}$ driving the atmospheric circulation [27], it is a clear possibility that large existing wind farms reach this limit. In this paper, an expression for the energy available for power production is derived from the mechanical energy budget of the ABL. This expression is fitted to data on actual power production by offshore wind in the North Sea. The fitted curve is used to find the expected power density of large wind farms and to find the power potential of the entire North Sea. Estimates on wake effects on power production are usually performed with complex numerical models [4,5,13]. Numerical models are highly important, but it can be difficult to extract physical understanding from complex numerical models. The analytical theory derived in this paper isolates first-order processes and provides understanding of the physics controlling the power production of offshore wind. The theory agrees with observations and the results of numerical models and, therefore, deepens the insight into the main physics of the problem.

This paper is structured as follows. Data on power density, capacity density, and capacity factors for existing wind farms in the North Sea are presented first. Then, a theoretical expression for the available energy is derived and fitted to the observed power productions. From this, the expected area needs for future wind farms in the North Sea are derived.

## Power production in the North Sea

The average capacity factors for 31 wind farms in the North Sea are calculated from half-hourly production data. Power production data are retrieved from Elexon [38], while the areas and installed capacities of the wind farms are retrieved from 4C Offshore [39]. The locations of the wind farms are shown in Fig 1. The total installed capacity, area, average capacity factors and averaging period for all wind farms included in this study are shown in S1 Table.

This study examines the limit to power production imposed by the input of energy from the atmosphere. This requires data from fully operational wind farms. The power production time series for all wind farms in the dataset are visually inspected and compared to the installed capacity to reveal periods where technical issues are reducing the capacity factors.

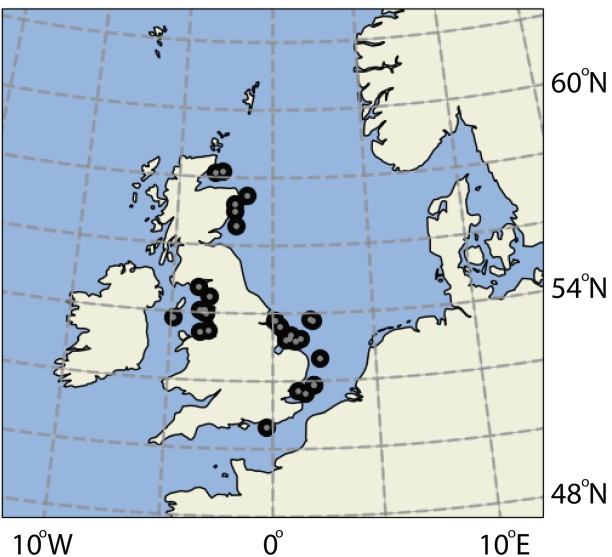

**Fig 1. The location of wind farms included in this study.** The Figure is made using the Cartopy Python package.

The capacity factors are determined by averaging over time periods in which the wind farms are running at full capacity. These are the averaging periods shown in the S1 Table. Four wind farms are excluded from the analysis because it was not possible to find a period where they were running at full capacity long enough to estimate a representative capacity factor. These wind farms are Seagreen, Moray East, Sheringham Shoals and Kincardine. The method of visual inspection is explained in S1-Appendix.

For some of the newest wind farms, data from 2024 is included in the averaging. The year 2024 includes data from January 1 to October 20. Thus, 2024 does not cover the full seasonal cycle, and this may have an effect on the derived capacity factor. However, for wind farms where data from 2024 are included, the capacity factor for 2024 is not significantly different from the capacity factor for 2023, and therefore it is assumed that the capacity factor derived by including data from 2024 is representative.

The averaged power density, derived from the total installed capacity, the capacity factor and area, and the capacity density are shown in Fig 2. The power densities vary from about 1 $Wm^{-2}$ to about 6 $Wm^{-2}$, and the capacity densities vary from about 3 $Wm^{-2}$ to 15 $Wm^{-2}$. Wind farms with the smallest areas cover the entire range of power and capacity densities, but the highest power and capacity densities for each area clearly decrease with increasing area. The capacity factors range from about 30 % to 53 %, and decrease as capacity densities increase (Fig 3).

The work presented in this paper is based on the hypothesis that the reduction of power densities with area can be explained by the energy input from the atmosphere. The next section investigates this further by analyzing the mechanical energy budget.

## Available kinetic energy for wind farm power production

In this section, an expression is derived for the energy available for power production. A basic assumption is that the vertical turbulent transfer of kinetic energy within the ABL makes the kinetic energy within the entire ABL available for power production, even when the height of the wind farm is only a fraction of the height of the ABL. The starting point of the analysis is

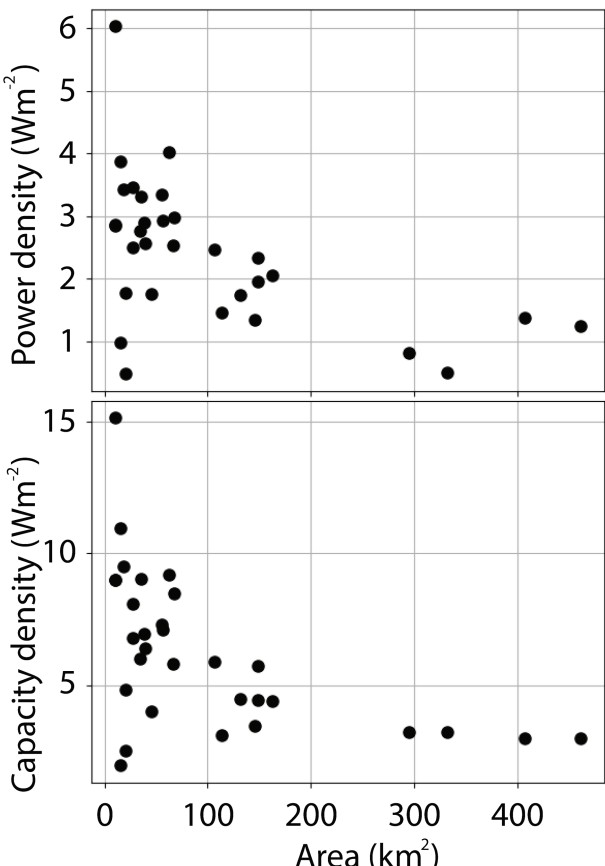

**Fig 2. a) Power density and b) capacity density for the wind farms presented in Fig 1 and S1 Table, plotted against wind farm area.**

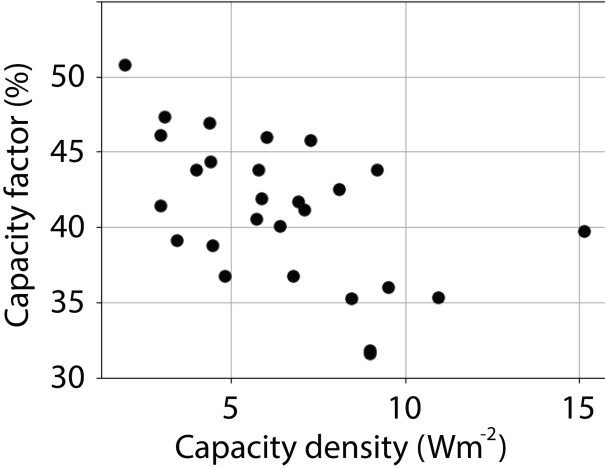

**Fig 3. Capacity factors for the wind farms presented in S1 Table, plottet against capacity density.** Capacity factors for Seagreen, Moray East, Sheringham Shoals and Kincardine wind farms are not shown.

the atmospheric momentum equations using pressure as vertical coordinate,

$$\frac{\partial u}{\partial t} + u\frac{\partial u}{\partial x} + v\frac{\partial u}{\partial y} + \omega\frac{\partial u}{\partial p} - fv = -g\frac{\partial z}{\partial x} + F_x \tag{1}$$

$$\frac{\partial v}{\partial t} + u\frac{\partial v}{\partial x} + v\frac{\partial v}{\partial y} + \omega\frac{\partial v}{\partial p} + fu = -g\frac{\partial z}{\partial y} + F_y. \tag{2}$$

$x$ and $y$ are coordinates in the north-south and east-west directions, $p$ is pressure, $z$ is geopotential height, $u$ and $v$ are horizontal velocities in eastward and northward direction, $\omega$ is vertical velocity in pressure coordinates, and $F_x$ and $F_y$ represents the frictional forces in $x$ and $y$ directions. In this coordinate system, the continuity equation reduces to,

$$\frac{\partial u}{\partial x} + \frac{\partial v}{\partial y} + \frac{\partial \omega}{\partial p} = 0. \tag{3}$$

Multiplying Eq. 1 by $u$ and Eq. 2 by $v$, adding together and using Eq. 3 gives the following equation for kinetic energy,

$$\frac{\partial K}{\partial t} + \frac{\partial(uK)}{\partial x} + \frac{\partial(vK)}{\partial y} + \frac{\partial(\omega K)}{\partial p} = -g\vec{v}\cdot\nabla_h z + uF_x + vF_y, \tag{4}$$

where

$$K = \frac{1}{2}\left(u^2 + v^2\right) \tag{5}$$

is the kinetic energy per mass. The term $-g\vec{v}\cdot\nabla_h z$ represents the conversion from potential to kinetic energy, which is the source of kinetic energy in the system. $\nabla_h$ is the horizontal gradient operator. Integrating over a mass unit $dm$, gives the kinetic energy balance. In pressure coordinates $dm$ is given by

$$dm = -\frac{1}{g}dxdydp. \tag{6}$$

To estimate the amount of energy available to wind power, Eq. 4 is first integrated from the surface to a specified height $H$. The energy budget for this surface layer can be expressed as

$$-\frac{1}{g}\left(\frac{\partial}{\partial t}\int_{p_s}^{p_H} Kdp + \frac{\partial}{\partial x}\int_{p_s}^{p_H} uKdp + \frac{\partial}{\partial y}\int_{p_s}^{p_H} vKdp + \omega_* K_h\right) = $$
$$\int_{p_s}^{p_H} \vec{v}\cdot\nabla_h z dp - \frac{1}{g}\int_{p_s}^{p_H}\left(uF_x + vF_y\right)dp. \tag{7}$$

To derive this equation, the following expression for the vertical velocity at a certain height $h$ is used.

$$\omega_h = \frac{dp_h}{dt} = \frac{\partial p_h}{\partial t} + u_h\frac{\partial p_h}{\partial x} + v_h\frac{\partial p_h}{\partial y} + \omega_*, \tag{8}$$

where the subscript $h$ represents the surface $p = p_h$, and $\omega_*$ is a turbulent entrainment velocity representing the vertical turbulent flux of kinetic energy at $p = p_h$. If the integration is performed to a height just above the turbines, $\omega_*$ will be a vital component of the flux of kinetic

energy into the wind farm [24]. Therefore, the winds above the wind farm are a reservoir of kinetic energy that is available to the wind farm. To include this in the analysis, the integration is performed to the top of the ABL. Here, frictional forces and turbulence are negligible [27], and in the continuation, $\omega_*$ is set equal to zero.

The next step is to integrate over the area of a wind farm to find an equation for the energy input to the farm.

$$-\frac{1}{g}\iint_A \left(\frac{\partial}{\partial t}\int_{p_s}^{p_H} K dp\right)dxdy - \frac{1}{g}\oint_C \left(\int_{p_s}^{p_H}\vec{v}K dp\right)\cdot\vec{n}dl$$

$$= \iint_A \left(\int_{p_s}^{p_H}\vec{v}\cdot\nabla_h z dp\right)dxdy \qquad (9)$$

$$-\frac{1}{g}\iint_A \left(\int_{p_s}^{p_H}(uF_x + vF_y)dp\right)dxdy.$$

$A$ is the area, $C$ is the circumference of the wind farm, $dl$ is an incremental distance along $C$ and $\vec{n}$ is the outward normal unit vector to $C$. The second term on the lhs is obtained using Green's theorem to transform the area integral of the divergence of $\vec{v}K$ to a line integral of the outward flux of $K$ across $C$.

The topic of this paper is the time-averaged power production of a wind farm. When averaging in time, the term containing the time derivative in Eq. 9 disappears and the equation reduces to

$$-\frac{1}{g}\oint_C \left\langle\left(\int_{p_s}^{p_H}\vec{v}K dp\right)\cdot\vec{n}\right\rangle dl - \iint_A \left\langle\left(\int_{p_s}^{p_H}\vec{v}\cdot\nabla_h z dp\right)\right\rangle dxdy$$

$$= -\frac{1}{g}\iint_A \left\langle\left(\int_{p_s}^{p_H}(uF_x + vF_y)dp\right)\right\rangle dxdy, \qquad (10)$$

where <> represents time averaging.

The sources of kinetic energy available for power production in the wind farm are represented in the lhs of Eq. 10. This includes the kinetic energy above the wind farm which is available through turbulent redistribution of the kinetic energy within the ABL. The term in the rhs of Eq. 10 represents the work done by frictional forces and contains the dissipation of kinetic energy. When a wind farm is present, the rhs must also contain a term representing the power production. In this case, the mechanical energy budget can be written,

$$-\frac{1}{g}\oint_C \left\langle\left(\int_{p_s}^{p_H}\vec{v}K dp\right)\cdot\vec{n}\right\rangle dl$$

$$-\iint_A \left\langle\left(\int_{p_s}^{p_H}\vec{v}\cdot\nabla_h z dp\right)\right\rangle dxdy = D + P, \qquad (11)$$

where $D$ is dissipation of kinetic energy and $P$ is electric power production. Eq. 11 can be written

$$W\frac{C}{A} + Z = \frac{D+P}{A}, \qquad (12)$$

where

$$W = -\frac{1}{gC} \oint_C \left\langle \left( \int_{p_s}^{p_H} \vec{v} K dp \right) \cdot \vec{n} \right\rangle dl, \tag{13}$$

and

$$Z = -\frac{1}{A} \iint_A \left\langle \left( \int_{p_s}^{p_H} \vec{v} \cdot \nabla_h z dp \right) \right\rangle dxdy. \tag{14}$$

$W$ is the advection of kinetic energy into the wind farm area by horizontal winds, averaged over the circumference of the wind farm, while $Z$ is the area averaged conversion from potential to kinetic energy within the wind farm. In Fig 4, $W$ is represented by the blue arrows illustrating the fluxes across the wind farm circumference, and $Z$ is represented by the broad green arrow illustrating the conversion from potential to kinetic energy within the wind farm. Eq. 12 clearly shows that the input of energy per area will decrease with increasing wind farm area, as the factor $C/A$ decreases with increasing area and it approaches zero when the area approach infinity.

## Comparing theory with actual power production

The wind farms represented in the dataset (S1 Table) have surface areas ranging from 10 km² to more than 400 km². Assuming that the wind farms are square in shape, the factor $C/A$ for the 10 km² wind farm is 6 times greater than for the 400 km² wind farm. At the time of writing, the Doggerbank wind farm is under construction and its total area is more than 1000 km². Thus, it is relevant to assume that $C/A$ varies with an order of magnitude over existing wind farm areas and that the solution to Eq. 12 will vary strongly with wind farm area. Both $W$ (Eq. 13) and $Z$ (Eq. 14) depend on the atmospheric conditions at the wind farm sites.

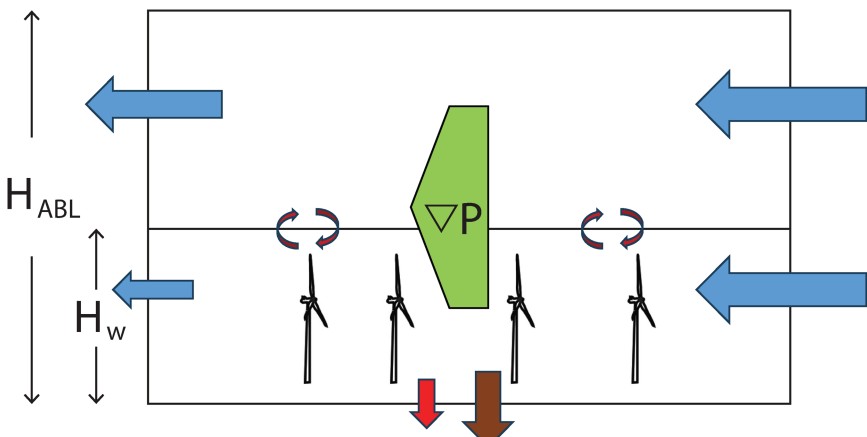

**Fig 4. Sketch illustrating the mechanical energy balance of the ABL.** The blue arrows represent the kinetic energy transported into the wind farm by incoming winds. The broad green arrow represents conversion from potential to kinetic energy and the brown and red arrows represent dissipation and power production. The kinetic energy within the ABL is defined as available energy for power production. Kinetic energy above the wind farm is made available to the wind farm by vertical turbulent exchange illustrated by the mixing arrows above the turbines. $H_w$ is the height of the wind farm and $H_{ABL}$ is the height of the ABL.

However, since $W$ depends on an average along the circumference of the wind farm and $Z$ on an average over the wind farm area, it is unlikely that they will vary significantly with area. Wind conditions in the area of interest are probably fairly equal, and it is reasonable to assume that the variability of $W$ and $Z$ within the dataset varies much less than the factor $C/A$, and that simply using the averages of $W$ and $Z$ will give reasonable results. This results in the following.

$$\overline{W}\frac{C}{A} + \overline{Z} = \frac{D+P}{A}, \tag{15}$$

where $\overline{W}$ and $\overline{Z}$ represents averages over all wind farms in the dataset. Eq. 15 describes the total energy input from the atmosphere and the question is how this relates to the power production. Rearranging Eq. 15 gives an expression for the power density

$$\frac{P}{A} = \frac{1}{1 + \frac{D}{P}}\left(\overline{W}\frac{C}{A} + \overline{Z}\right). \tag{16}$$

The power density depends on the relation between $P$ and $D$. To explore this relation in more detail, assume a wind farm, covering a specific surface area (say 100 km²), where the installed capacity is increased from zero up to a level where the power production is limited by the input of energy from the atmosphere. When the installed capacity is zero, the input of energy is balanced by dissipation only. For zero installed capacity the total dissipation within the area is $D_0$, and $D/P$ is infinite. If one turbine is installed, assume power production is $\delta P$ and dissipation increases with $\delta D$. $\overline{W}$ will increase by $\delta \overline{W}$ because the turbine damp the wind leading to a reduction of the kinetic energy flux out of the area compared to the influx, thus increasing the integral in Eq. 13. However, one turbine will have very little effect on the wind within the specific area, and the power production is very small compared to the available energy. Therefore, $\delta D << D_0$, and $D/P \sim D_0/\delta P$. If a second turbine is installed, the effect on the wind will still be small, but the power production will double. Thus, now $D/P \sim D_0/2\delta P$. This suggests that if the power production is lower than the limit set by the input of energy from the atmosphere, $D/P$ will decrease with an increase in installed capacity. However, there is a limit to the flux of energy into the wind farm, and when $P+D$ reach this limit a further increase in the installed capacity will lead to $P+D$ being unchanged or even reduced. In a study of tidal power, Garrett and Cummins [40] find the power potential by increasing the installed capacity in their model until power production reaches a maximum where further increase in the capacity reduces power production. Power production and dissipation in wind farms is likely to behave in the same way. Thus, if we assume the wind farms in the North Sea is limited by atmospheric energy input, $P+D$ will be independent on the installed capacity. This does not necessarily mean that $D/P$ is constant, but it seems reasonable that the values of $D/P$ will be approximately equal for wind farms in the North Sea that is limited by the atmospheric energy input. Therefore, in the continuation of this analyses it is assumed that the main variations in power density is caused by the variations in $C/A$ and $D/P$ can be treated as a constant. However, the analyses of $D/P$ suggest that only within wind farms operating at the limit given by the atmospheric energy input can the power density be expected to vary according to the variations in $C/A$. To a first approximation, the power density $P/A$ will now be given as

$$\frac{P}{A} = W_P\frac{C}{A} + Z_P, \tag{17}$$

where

$$W_P = \frac{\overline{W}}{1 + \frac{D}{P}}$$

$$Z_P = \frac{\overline{Z}}{1 + \frac{D}{P}}.$$

(18)

To arrive at Eq. 17, the basic assumption is that the variables that depend on the average meteorological conditions and the fraction $P/D$ can be treated as constants in relation to the large variations in $C/A$. The assumption that $P/D$ vary slowly in comparison with $C/A$ requires that wind farm power production is close to the atmospheric limit, such that an increase in installed capacity will not lead to an increase in power production.

Wind farms that are limited by the atmospheric energy input should, according to Eq. 17, be a linear function of $C/A$. In the following analysis, the four wind farms that were found to not operate at full capacity are left out. The dataset used in the following analysis now includes data from 27 wind farms. Fitting Eq. 17 to the actual power production for all wind farms in the dataset gives the result shown in Fig 5a. The fit has a correlation coefficient of 0.52. The fitting is done by assuming that all wind farms are shaped as squares such that $C/A = 4\sqrt{1/A}$. An analysis of the errors introduced by this assumption is given in the Discussion section. For wind farms with areas of 50 km² or less, the power density varies from 1 to about 6 MW/km². Of these wind farms, those with the lowest power density are probably not limited by the atmospheric energy input and therefore cannot be expected to fit well with the expression in Eq. 17. Wind farms with the lowest power density are those with the lowest installed capacity, and Fig 5b shows the fit to the actual power density when wind farms with installed capacity lower than 100 MW are removed. The correlation coefficient now increases to 0.84. There are 23 wind farms with installed capacity larger than 100 MW. Removing wind farms with installed capacity less than 200 MW leads to the fit shown in Fig 5c, which has a correlation coefficient of 0.77. There are 18 wind farms with installed capacity larger than 200 MW.

Fig 5 shows a good correlation between the functional form given by Eq. 17 and the actual power densities, when the smallest wind farms are removed. In an expansion of offshore wind in the North Sea, the value of $Z_P$ is especially interesting because it determines the upper limit of power densities in very large wind farms. Removing wind farms with installed capacity below 100 MW and below 200 MW (Fig 5) gives $Z_P$ equal to 0.78 $Wm^{-2}$ and 0.74 $Wm^{-2}$, and $W_P$ equal to 3414 $Wm^{-1}$ and 3729 $Wm^{-1}$. A method to estimate uncertainty is to calculate the difference between the actual power density and the theoretical power density given by Eq. 17. This is done as follows.

$$\delta p = \sqrt{\frac{(pa - pt)^2}{N}},$$

(19)

where $pa$ is the observed power density, $pt$ is the theoretical power density given by Eq. 17, and $N$ is the number of wind farms. $\delta p$ given by Eq. 19 is independent of $C/A$ and can therefore be added to $Z_P$ without influencing $W_P$. However, since it is derived from the difference between the actual power density and the power density derived from Eq. 17, uncertainties in the term containing $W_P$ also contribute to the value of $\delta p$. The power densities of wind farms with a capacity greater than 100 MW (Fig 5b) is chosen to estimate uncertainties using Eq. 19. This dataset has $N = 23$ and gives $\delta p = 0.58\,Wm^{-2}$, again giving $Z_P = 0.78 \pm 0.58$. The reason for using this dataset is that it gives a fairly similar result to using data from wind farms with installed capacity greater than 200 MW (Fig 5c), but shows a higher correlation to the theory and contains data from more wind farms.

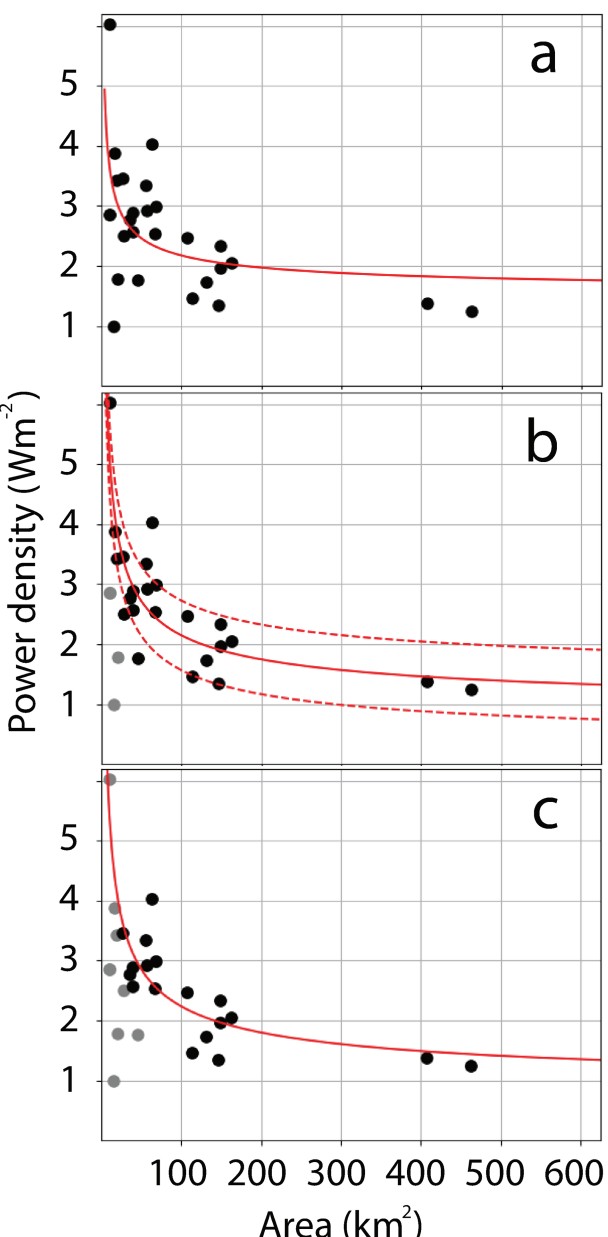

**Fig 5. Results of Eq. 17 fitted to observed power densities.** Black dots represent observed power density and the red line the best fit of Eq. 17 to the data. a) Eq. 17 fitted to all data (27 data points). b) Eq. 17 fitted to data from wind farms with capacity larger than 100 MW (23 data points). Data from wind farms with capacity lower than 100 MW is shown by grey dots. c) Eq. 17 fitted to data from wind farms with capacity larger than 200 MW (18 data points). Data from wind farms with capacity lower than 200 MW are shown by grey dots. The dotted red lines in b) marks the upper and lower limit based on the uncertainty estimated by Eq 19.

## Discussion

The results presented in this paper show that the production of electricity from large offshore wind farms in the North Sea follows the same functional dependency as the available kinetic energy given by Eq. 12. This suggests that power production is limited by the input of energy

from the atmosphere. The basic concept of the theory presented is that the kinetic energy within the entire ABL is available for power production. The dynamics considered is in agreement with the work of Antonini and Caldeira [26,28]. They compare the flow inside a wind farm with an Ekman layer, where the friction or resistance caused by wind turbines leads to ageostrophic flow towards lower pressure. In an Ekman layer, it is the conversion from potential to kinetic energy that balances the loss of kinetic energy due to friction. $Z$ (Eq. 14) can be understood as the conversion from potential to kinetic energy as a result of flow resistance caused by friction and power production, while $W$ (Eq. 13) represents the kinetic energy advected into the wind farm by horizontal winds. The power density is shown to be given by a linear function of $C/A$ with coefficients $W_P$ and $Z_P$ (Eq. 17). $W_P$ and $Z_P$ depend on atmospheric conditions through average values of $W$ and $Z$, and the relationship between dissipation and power production ($D/P$, see Eq. 18). In the derivation of the power density, $W_P$ and $Z_P$ are treated as constants. This is based on two assumptions, where the first is that $W$ and $Z$ can be replaced by their averages over the North Sea region. This is okay as long as the variability in power density caused by $C/A$ is much larger than the variability caused by variations in $W$ and $Z$. The second assumption is that $D/P$ can also be treated as a constant. As argued in connection with Eq. 16, this is okay if the wind farm operates at the limit set by the atmospheric energy input. However, if the wind farm operates below this limit, $D/P$ will depend on the installed capacity. To be limited by the atmospheric energy input, a capacity large enough to exploit the available energy needs to be installed. Therefore, it is the largest wind farms that are most likely to reach this limit. Fig 5 shows that the power density in the largest wind farms agrees with the functional relationship given by Eq. 17, while wind farms with installed capacity less than 100 MW show a poor fit. A likely explanation is that the smallest wind farms are not limited by the input of energy from the atmosphere and do not fit the function given by Eq. 17 because the assumption that $D/P$ is constant does not hold for these wind farms. Therefore, Eq. 17 is only valid for wind farms with power production limited by atmospheric energy input, and the fact that the small wind farms do not fit can be seen as a support of the theory.

Another uncertainty comes from the assumed shape of the wind farms. When fitting the theory to the actual power densities, $C/A$ is determined by assuming that all wind farms are square-shaped. This is of course an approximation as wind farms are usually not shaped as perfect squares. However, what is most important is that $C/A$ can be approximated as $Q/\sqrt{A}$, where $Q$ is a constant. This is the case for most shapes, $Q = 4$ for a square, $Q = 2\sqrt{\pi}$ for a circle, and $Q = 3\sqrt{2}$ for a rectangle where the shortest side is half the length of the longest side. The values of $W_P$ and $Z_P$ are found by fitting the data to a function of the form $W_P \cdot x + Z_P$, where $x = 4/\sqrt{A}$, which represents $C/A$ for a square. $P/A$ and $Z_P$, as given by Eq. 17, are not affected by the choice of shape. Therefore, the plots shown in Fig 5 are not affected, while $W_P$ is affected because $C/A$ depends on the shape of the wind farm. What adds to the uncertainty is that the wind farms cannot all be represented by one shape and that the variability in the shapes leads to deviations in the way $C/A$ relates to $1/\sqrt{A}$. An idea of the size of the deviations can be obtained by comparing the power density obtained from Eq. 17 by using $C/A$ from different shapes with equal area and equal values of $W_P$. Consider three shapes, a square, a circle, and a rectangle, where the shortest sides are half the length of the longest. The largest difference in power density for these three shapes, using $W_P = 3414$, is 0.75 $Wm^{-2}$ for $A = 10 km^2$, 0.33 $Wm^{-2}$ for $A = 50 km^2$, 0.23 $Wm^{-2}$ for $A = 100 km^2$, 0.17 $Wm^{-2}$ for $A = 200 km^2$, 0.13 $Wm^{-2}$ for $A = 300 km^2$, 0.12 $Wm^{-2}$ for $A = 400 km^2$ and 0.11 $Wm^{-2}$ for $A = 500 km^2$. For wind farms smaller than 50 $km^2$, this difference is similar or greater than the uncertainty given by Eq. 19. For an increasing area, the difference in power density between the three shapes will tend to zero, and for wind farms larger than 50 $km^2$ the difference is less than the uncertainty given

by Eq. 19. However, when discussing wind farm shapes, the orientation of the wind farm relative to the main wind direction is important. An investigation of this is beyond the scope of this study, but the high correlation between theory and data is a strong indicator that the idealized theory presented includes the main physics of the problem.

Another important topic to discuss is the installed capacity density, which has a similar variability to that of the power density (Fig 2). Does this mean that the observed dependency of power density with area is simply a result of how the capacity is installed and has little to do with atmospheric available energy? This is unlikely for the following reasons. For companies that install and drift a wind farm, it is important to have as high capacity factors as possible. Low capacity factors are bad economy. An extensive experience of wind farm power production and its variation with area is obtained from the many wind farms that is installed in the North Sea. In addition, the efficiency of the turbine layout within a wind farm is carefully analyzed prior to the installation of the wind farm using models and data from existing wind farms in the area. In doing this, they will find that installing the same total capacity within a smaller area will lead to lower capacity factors, and thus to lower power production. This is supported by the dependency of capacity factors on capacity density, which indicates that higher capacity density leads to lower capacity factors (Fig 3). Thus, if the available energy is the limit to power production, it is natural that the installed capacity mirrors the available energy. Therefore, the fit between observations and theory presented in Fig 5 strongly indicates that the production of offshore wind power in the North Sea is limited by atmospheric energy input.

The values of $W_P$ and $Z_P$ are determined by fitting with Eq. 17, and these values can now be used to predict power generation in future wind farms within the North Sea. Since the available energy is defined as the kinetic energy within the full height of the ABL, the amount of available energy that is converted to electrical power depends on the turbulent vertical transfer that transports the kinetic energy from the upper part of the ABL to the wind farm below. Thus, although vertical turbulent exchange is not explicitly treated, the processes controlling vertical exchange, such as atmospheric stability, are embedded in variables $W_P$ and $Z_P$ (Eq. 17). These variables determine how much of the available energy is converted to electric power.

The results presented in Table 1 are obtained by fitting Eq. 17 to the power production of wind farms with installed capacity of 100 MW or greater (Fig 5 b). For $Z_P = 0.78\,Wm^{-2}$, the energy carried by the incoming winds contributes the main share of the kinetic energy to the

**Table 1. Predicted power production for future wind farms.**

| Area ($km^2$) | $W_P \frac{C}{A}$ ($Wm^{-2}$) | PD ($Wm^{-2}$) | PG (MW) | APG (TWh) |
|---|---|---|---|---|
| 10 | 4.30 | 5.10 (0.58) | 51.0 (5.8) | 0.45 (0.05) |
| 50 | 1.93 | 2.71 (0.58) | 135.5 (29.0) | 1.19 (0.25) |
| 100 | 1.37 | 2.15 (0.58) | 215 (58) | 1.88 (0.5) |
| 400 | 0.68 | 1.46 (0.58) | 584 (232) | 5.1 (2.0) |
| 1000 | 0.43 | 1.21 (0.58) | 1210 (580) | 10.6 (5.1) |
| 4000 | 0.22 | 1.00 (0.58) | 4000 (2320) | 35.0 (20.3) |
| 10000 | 0.14 | 0.92 (0.58) | 9200 (5800) | 80.6 (50.1) |
| 100000 | 0.04 | 0.82 (0.58) | 82000 (58000) | 718 (508) |
| 625000 | 0.02 | 0.80 (0.58) | 500000 (362500) | 4380 (3175) |

Area (km²), $W_P \frac{C}{A}$ ($Wm^{-2}$), predicted power density, PD ($Wm^{-2}$), predicted power generation, PG (MW) and annual power generation, APG (TWh). The predictions are based on the use of $W_P = 3414\ Wm^{-1}$ and $Z_P = 0.78 \pm 0.58\ Wm^{-2}$ in Eq. 17. The values in parenthesis represents the uncertainties.

wind farm up to a wind farm size of approximately 400 $km^2$. For a wind farm with an area of 4000 $km^2$, incoming winds contribute about a quarter of the energy input. From the results presented in Table 1, it can be expected that a wind farm with an area equal to 100000 $km^2$ produces between 718 ± 508 TWh per year and about 5 % of the energy input is carried by incoming winds. The remaining 95 % comes from the conversion from potential to kinetic energy within the wind farm.

It is interesting to see how the predicted power production in very large wind farms ($Z_P$) compares to the actual conversion from potential to kinetic energy in a reanalysis dataset. The conversion from potential to kinetic energy in the lower 1000 m estimated from the NCEP-DOE reanalysis data [41] is shown in Fig 6. This varies from about 2.75 $Wm^{-2}$ in the south to approximately 3.75 $Wm^{-2}$ in the northern part of the North Sea. The value of $\overline{Z}$ is then approximately 3 $Wm^{-2}$, assuming that a height of 1000 m represents the height of the ABL, and that the conversion from potential to kinetic energy does not change significantly in the presence of a wind farm. $Z_P = 0.78\,Wm^{-2}$ and $\overline{Z} = 3\,Wm^{-2}$ gives $P/D = 0.26$. Including the uncertainty of $0.58\,Wm^{-2}$, gives $P/D$ ranging from about 0.1 to 0.5. This seems reasonable considering that $D$ represents the total dissipation of energy in the ABL, and that wind farms cover only a fraction of the ABL. Miller and Kleidon [21] simulated a global wind farm that produces about $\sim 0.55\ Wm^{-2}$ over the ocean, compared to a total dissipation of $\sim 2.2\ Wm^{-2}$. This gives $P/D \sim 0.25$, which agrees with the estimate presented above. Although these are fairly crude estimates, they confirm that the estimated values of $Z_P$ are not contradicted by the analysis of the atmospheric reanalysis data.

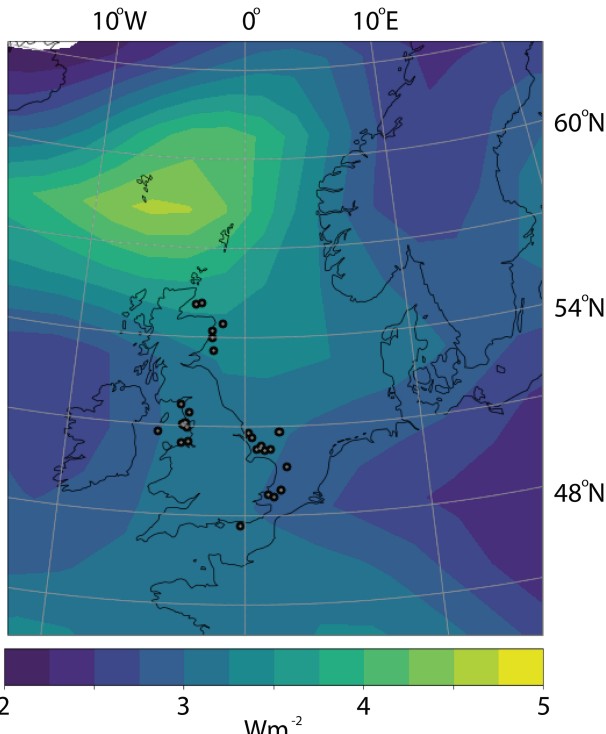

**Fig 6. $\bar{v} \cdot \nabla_h z$ integrated from the surface to 1000 m height and averaged over the years 2020, 2021 and 2022 using NCEP-DOE reanalysis data [41].** The Figure is made using the Cartopy Python package.

How large an area will a scale-up of offshore wind in the North Sea require? The last row in Table 1 gives the power production of an area that represents the entire North Sea and the Irish Sea. A wind farm covering this area will produce about 500 ± 363 GW, which corresponds to an annual power production of about 4400 ± 3200 TWh. To obtain 50% capacity factor, this will fit an installed capacity between 300 and 1700 GW. However, it may be more efficient to install several smaller wind farms if there is enough area to place them far enough away from each other to not interact. 330 wind farms, the same size as Hornsea 1 (see S1 Table), equal 400 GW of installed capacity. This will produce 200 GW with a capacity factor of 50%, and require a combined area of 135500 $km^2$. This is about a fifth of the entire area of the North Sea plus the Irish Sea, and it is questionable if it is possible to install those many 1.2 GW wind farms in this area without interaction between them, which will reduce capacity factors. A production of 200 GW will give about 1750 TWh per year, which is close to half of the European electricity production today.

Assessment of wind resources often uses simplistic methods to account for wake losses [15]. A common method in resource assessments is to compute capacity factors from atlases of undisturbed wind. Rapid reduction in power density with area is caused by the damping of the wind within a wind farm. Undisturbed wind is a good basis for estimating the power production of small wind farms, but is less relevant for the power production of large wind farms. The results presented here suggest that it is the mean conversion from potential to kinetic energy that supplies the large wind farms with energy. This is given by the mean pressure gradients, in line with the results of Antonini and Caldeira [26]. Strong undisturbed winds may, of course, be the result of strong pressure gradients, and as such, the wind speed is also a measure of the pressure gradients. However, strong wind may result from low surface roughness and a medium-strong pressure gradient. In the last case, the energy supply to very large wind farms may not be that large. Thus, for resource assessments of very large wind farms, the mean pressure gradients in the area may be better suited than the undisturbed wind. A power density estimated as a function of the area and circumference of wind farms (Eq. 17) can be a road to better estimates of resource potentials.

## Conclusion

This article develops a theory for the production of electricity by offshore wind farms in the North Sea based on available energy in the atmosphere. A comparison of theory with actual power production by 27 existing wind farms shows that the power production in about 23 of the largest wind farms is limited by the input of energy from the atmosphere. The power density of these wind farms is described by a linear function of $C/A$, where $C$ is the circumference and $A$ is the area of the wind farm. This function is fitted to the power production of existing wind farms and is used to predict the power production and area usage of future wind farms. Wind farms with an area less than 10 $km^2$ may have power densities larger than 6 $Wm^{-2}$. The power density is reduced with increasing areas, and the largest wind farms today, with areas just over 400 $km^2$ have power densities just above 1 $Wm^{-2}$. For larger future wind farms, the power densities will asymptotically approach a constant value of 0.78 ± 0.58 $Wm^{-2}$ when the area approaches infinity. This result is in agreement with other studies on the potential of offshore wind power based on an atmospheric mechanical energy budget [21,22].

Since available energy is the limiting factor for power production, reliable estimates of power potential must include an atmospheric energy budget. A scale-up of offshore wind in the North Sea, in the order of a few hundred GW, will reduce area efficiency and lead to power densities near or below 1 $Wm^{-2}$.

## Supporting information

**S1 Table. Wind farms in the North Sea.** Total installed capacity (Cap), area, average capacity factors, and averaging period in the dataset analyzed in this paper.
(PDF)

**S1 Appendix. Identifying wind farms not operating at full capacity.** A description of the method to identify wind farms that do not operate at full capacity.
(PDF)

## Acknowledgments

Thanks to the three anonymous reviewers whose constructive comments helped to improve the manuscript. NCEP/DOE Reanalysis II data provided by the NOAA PSL, Boulder, Colorado, USA, from their website at https://psl.noaa.gov. Contains BMRS data © Elexon Limited copyright and database right [2024].

## Author contributions

**Conceptualization:** Ole Anders Nøst.

**Data curation:** Ole Anders Nøst.

**Formal analysis:** Ole Anders Nøst.

**Funding acquisition:** Ole Anders Nøst.

**Investigation:** Ole Anders Nøst.

**Methodology:** Ole Anders Nøst.

**Validation:** Ole Anders Nøst.

**Visualization:** Ole Anders Nøst.

**Writing – original draft:** Ole Anders Nøst.

**Writing – review & editing:** Ole Anders Nøst.

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
