## [Decision Letter · Decision Letter 0]

30 Aug 2024

PONE-D-24-31465Power production and area usage of offshore wind and the relationship with available energy in the atmospherePLOS ONE

Dear Dr. Nøst,

Thank you for submitting your manuscript to PLOS ONE. After careful consideration, we feel that it has merit but does not fully meet PLOS ONE’s publication criteria as it currently stands. Therefore, we invite you to submit a revised version of the manuscript that addresses the points raised during the review process. Please address the reviewers' comments carefully and submit the revised paper and response letter according to the requirements, as given below. 

We look forward to receiving your revised manuscript.

Kind regards,

Mohamad Abou Houran

Academic Editor

PLOS ONE

“This work has been funded by the Norwegian Seafood Research Fund grant no. 901916.”

“This study was supported by the Norwegian Seafood Research Fund (https://www.fhf.no/) grant number 901916. The funders had no role in study design, data collection and analysis, decision to publish, or preparation of the manuscript.”

3. We note that Figures 1 and 6 in your submission contain [map/satellite] images which may be copyrighted. All PLOS content is published under the Creative Commons Attribution License (CC BY 4.0), which means that the manuscript, images, and Supporting Information files will be freely available online, and any third party is permitted to access, download, copy, distribute, and use these materials in any way, even commercially, with proper attribution. For these reasons, we cannot publish previously copyrighted maps or satellite images created using proprietary data, such as Google software (Google Maps, Street View, and Earth). For more information, see our copyright guidelines: http://journals.plos.org/plosone/s/licenses-and-copyright.

1. You may seek permission from the original copyright holder of Figures 1 and 6 to publish the content specifically under the CC BY 4.0 license. 

Reviewers' comments:

Reviewer's Responses to Questions

**Comments to the Author**

1. Is the manuscript technically sound, and do the data support the conclusions?

Reviewer #1: Yes

Reviewer #2: Yes

2. Has the statistical analysis been performed appropriately and rigorously? 

Reviewer #1: Yes

Reviewer #2: Yes

3. Have the authors made all data underlying the findings in their manuscript fully available?

Reviewer #1: No

Reviewer #2: Yes

4. Is the manuscript presented in an intelligible fashion and written in standard English?

Reviewer #1: Yes

Reviewer #2: Yes

5. Review Comments to the Author

Reviewer #1: The authors have done an excellent job; however, here are my comments:

1) In Table 1, in the Dudgeon row under the Area column, there is an extra "N".

2) Please clarify which unit of measurement "WM^-2" refers to in Figure 2.

3) The conclusions repeat the same idea regarding the C/A ratio; please improve the conclusions.

4) I believe the paper should include a discussion section. Since this is a methodological proposal, it should have an extensive discussion. Please add this section.

Reviewer #2: The paper is well-written, and it seems of value for wind energy community both scientific and

industry. Please see comments below:

Small edits:

1. Line 22 p1/14: within 2050 → by 2050

2. future area “needs” for offshore wind. → “needed” for offshore wind.

3. Please give a simple and short definitions to capacity density, capacity factor for readers

outside of the field.

4. Table 1 caption mentions that the table has the coordinates and locations of the farms but

there is no such information on the table. This table can also be moved to supplementary

materials. You should also define cf202X meaning in the caption such as “Capacity

Factor for 202X (cf202X)”.

5. Please discuss the details of what is in figure 2 in caption. Figure 2 has parts a, and b.

You should have a brief explanation for these in the caption.

6. In equation 9, what is ndl?

7. You don’t explain what the mixing arrows are in the figure 4.

Please see comments below:

1. “However, since W depends on an average along the circumference of the wind farm and

Z on an average over the wind farm area, it is unlikely that they will vary significantly

with area.” → does this mean that you are ignoring the role of atmospheric stability in

your calculations?

2. Can you provide steps to how you rearranged 15 to 16?

3. “If the installed capacity is zero (no wind farm), Eq 15 still applies, but the input of

energy will be balanced by dissipation only. When increasing the installed capacity, the

power production will increase until it reaches a limit, given by the input of energy.” →

please clarify this by determining why these sentences are true. For instance, “when

increasing the installed capacity, the power production increase, [add what would change

in Eq 15], and therefore it reaches a limit.

4. Line 229, page 8: “assuming that all wind farms are shaped as squares”, this is a big

assumption as in reality the wind farms are hardly shaped as a perfect square. What

uncertainties do you think this assumption adds to your calculations? Please add a few

sentences discussing this.

5. In page 8, when you remove wind farms with smaller capacity (<200MW) you get better

correlation. Please consider explaining a reason for this. And specify how many of the

wind farms are getting removed by removing the small farms (what percent of the wind

farms are removed)?

6. In Discussion, you mention “Wind farms with a total capacity below 200 MW (shown in

gray in Fig 5 b and c) probably operate with a lower efficiency and therefore the

assumption of constant WP and ZP is not valid for these wind farms.” → How did you

decide on threshold of 200MW as for farms that do not perform with high efficiency?

7. Table 2, are you using ZP = 0.3 divided by 0.8? If so, why is that? On page 9 you

mentioned that it could vary between these two numbers as upper and lower limits.

6. PLOS authors have the option to publish the peer review history of their article (what does this mean?). If published, this will include your full peer review and any attached files.

Reviewer #1: No

Reviewer #2: No

---

## [Author Response · Author response to Decision Letter 1]

28 Nov 2024

I have prepared detailed responses to all reviewers and editor comments in the response to reviewer document

---

## [Decision Letter · Decision Letter 1]

24 Jan 2025

PONE-D-24-31465R1Power production and area usage of offshore wind and the relationship with available energy in the atmospherePLOS ONE

Dear Dr. Nøst,

Thank you for submitting your manuscript to PLOS ONE. After careful consideration, we feel that it has merit but does not fully meet PLOS ONE’s publication criteria as it currently stands. Therefore, we invite you to submit a revised version of the manuscript that addresses the points raised during the review process.

We look forward to receiving your revised manuscript.

Kind regards,

Mohamad Abou Houran

Academic Editor

PLOS ONE

Additional Editor Comments"

Dear authors,

Please address the reviewer's comments carefully and improve the paper.

Reviewers' comments:

Reviewer's Responses to Questions

**Comments to the Author**

1. If the authors have adequately addressed your comments raised in a previous round of review and you feel that this manuscript is now acceptable for publication, you may indicate that here to bypass the “Comments to the Author” section, enter your conflict of interest statement in the “Confidential to Editor” section, and submit your "Accept" recommendation.

Reviewer #1: All comments have been addressed

Reviewer #3: (No Response)

2. Is the manuscript technically sound, and do the data support the conclusions?

Reviewer #1: Yes

Reviewer #3: Partly

3. Has the statistical analysis been performed appropriately and rigorously? 

Reviewer #1: Yes

Reviewer #3: Yes

4. Have the authors made all data underlying the findings in their manuscript fully available?

Reviewer #1: Yes

Reviewer #3: Yes

5. Is the manuscript presented in an intelligible fashion and written in standard English?

Reviewer #1: Yes

Reviewer #3: Yes

6. Review Comments to the Author

Reviewer #1: The authors followed all the suggestions requested so, my recommendation is to accept the manuscript.

Reviewer #3: The paper should make some improvements:

1. The innovation is not clear in the beginning. The reason to publish it is that the added value of this work for the global S&T audience (introduction) needs more definition in the introduction.

2. The sustainable siting parameter beyond the wind potential should be included in the state-of the art (environment, regulations, other uses, etc). Several very recently published scientific works are available.

7. PLOS authors have the option to publish the peer review history of their article (what does this mean?). If published, this will include your full peer review and any attached files.

Reviewer #1: No

Reviewer #3: No

---

## [Author Response · Author response to Decision Letter 2]

13 Feb 2025

Reviewer #1: The authors followed all the suggestions requested so, my recommendation is to

accept the manuscript.

Answer: Thank you! I have worked hard to come up with solid responses to your comments and this

has clearly improved the quality of the manuscript.

Reviewer #3: The paper should make some improvements:

1. The innovation is not clear in the beginning. The reason to publish it is that the added value of

this work for the global S&T audience (introduction) needs more definition in the introduction.

Answer: The work presented in the manuscript derives a fundamental theory for the power

production of offshore wind. The theory is based on the governing equations for the

atmospheric momentum balance and arrives at a simple relation between power density

and the area of a wind farm. It is well known that power density decreases with increasing

area, but until now it has not been clear how large a wind farm must be for this process to

be significant. From the developed theory and its agreement with actual measured power

production, this is significant for existing wind farms ranging in size from almost zero to

more than 400 km2.

In addition, from the result one can conclude that the limiting factor for power production is

the input of energy from the atmosphere. There are many recent estimates of power

potential that do not consider the atmospheric energy budget. From the results presented

we now know that the atmospheric energy budget is vital. It is the limiting factor and

therefore must be part of a realistic estimate of power potential.

Finally, a fundamental analytical theory that agrees with data is important. It brings

understanding and insight into the problem.

I have added these sentences to the end of the introduction.

“Estimates on wake effects on power production are usually performed with complex

numerical models [4-6]. Numerical models are highly important, but it can be difficult to

extract physical understanding from complex numerical models. The analytical theory

derived in this paper isolates first-order processes and provides understanding of the

physics controlling the power production of offshore wind. The theory agrees with

observations and the results of numerical models and, therefore, deepens the insight into

the main physics of the problem.”

2. The sustainable siting parameter beyond the wind potential should be included in the state-of

the art (environment, regulations, other uses, etc). Several very recently published scientific

works are available.

Answer: This is good advice, and I have added about one page in the introduction to cover these

topics. I have focused on the environmental effects caused by wind wakes. This is natural

since the energy budget analysis in the manuscript is essentially a method to quantify the

effect of wakes. Wakes have effects on the ocean circulation and ecosystem and may also

cause conflicts between wind farm operators and countries. This is now described in the

introduction and several recent publications are included.

---

## [Editor Report · Decision Letter 2]

10 Mar 2025

Power production and area usage of offshore wind and the relationship with available energy in the atmosphere

PONE-D-24-31465R2

Dear Dr. Ole,

We’re pleased to inform you that your manuscript has been judged scientifically suitable for publication and will be formally accepted for publication once it meets all outstanding technical requirements.

Kind regards,

Mohamad Abou Houran

Academic Editor

PLOS ONE

Additional Editor Comments (optional):

The authors have addressed the reviewer's comments. The work can be published. 
---

## [Editor Report · Acceptance letter]

PONE-D-24-31465R2

PLOS ONE

Dear Dr. Nøst,

I'm pleased to inform you that your manuscript has been deemed suitable for publication in PLOS ONE. Congratulations! Your manuscript is now being handed over to our production team.

Kind regards,

on behalf of

Dr. Mohamad Abou Houran

Academic Editor

PLOS ONE